# Bioluminescent-Inhibition-Based Biosensor for Full-Profile Soil Contamination Assessment

**DOI:** 10.3390/bios12050353

**Published:** 2022-05-19

**Authors:** Elizaveta M. Kolosova, Oleg S. Sutormin, Aleksandr A. Shpedt, Ludmila V. Stepanova, Valentina A. Kratasyuk

**Affiliations:** 1Department of Biophysics, Siberian Federal University, Krasnoyarsk 660041, Russia; osutormin@sfu-kras.ru (O.S.S.); lstepanova@sfu-kras.ru (L.V.S.); vkratasyuk@sfu-kras.ru (V.A.K.); 2Federal Research Center ‘Krasnoyarsk Science Center Siberian Branch of the Russian Academy of Sciences’, Krasnoyarsk 660036, Russia; ashpedt@sfu-kras.ru; 3Department of Aquatic and Terrestrial Ecosystems, Siberian Federal University, Krasnoyarsk 660041, Russia; 4Photobiology Laboratory, Institute of Biophysics, Federal Research Center ‘Krasnoyarsk Science Center, Siberian Branch of the Russian Academy of Sciences’, Krasnoyarsk 660036, Russia

**Keywords:** bioluminescent biosensor, enzyme bioassay, soil pollution, soil profile

## Abstract

A bioluminescent-enzyme-inhibition-based assay was applied to predict the potential toxicity of the full profile of the following soil samples: agricultural grassland, 10-year fallow land (treated with remediation processes for 10 years) and uncontaminated (virgin) land. This assay specifically detects the influence of aqueous soil extracts from soils on the activity of a coupled enzyme system of luminescent bacteria: NAD(P)H:FMN-oxidoreductase + luciferase (Red + Luc). It was shown that the inhibitory effect of the full-profile soil samples on the Red + Luc system decreased with depth for the 10-year fallow-land and virgin-land samples, which correlated with a decrease in the humic organic matter content in the soils. The inhibitory effect of the agricultural grassland on the Red + Luc enzyme system activity was more complex and involved the presence of the humic organic matter content, as well as the presence of pollutants in the whole-soil profile. However, if the interfering effect of humic organic substances on the Red + Luc system’s activity is taken into account during full-profile soil toxicity assessments, it might help to detect pollutant mobility and its leaching into the subsoil layer. Thus, this bioluminescent method, due to the technical simplicity, rapid response time and high sensitivity, has the potential to be developed as a biological part of the inhibition-based assay and/or biosensors for the preventive tracing of potential full-profile soil contamination.

## 1. Introduction

Enzyme-based biosensors are analytical devices that use specific biochemical reactions mediated by isolated enzymes to detect chemical compounds, as a rule, by electrical, thermal or optical signals [1]. These biosensors must contain relevant biological parts and transducers for a forthcoming bioassay. In this case, biosensors have vital factors, such as suitability to be used in a field with harsh conditions and low costs for manufacturing, miniaturization and saving of materials. Qualitative biosensors and bioassays have the potential of being used for initial investigations of testing water, soil and air contamination caused by man-made loads.

A variety of enzymatic bioassays are widely used as research tools for the bioassay of water samples [2]. Nevertheless, there is currently a dire need for the diversification of bioassays and biosensor applications to assess the contamination of matrices more complex than water samples, for example, soil samples [3]. At the same time, a vast number of bioassay methods are available for soil pollution assessment and environmental monitoring, but little is known about whether they can be used to monitor full-profile soil toxicity. For example, only topsoil samples are collected when earthworms are used as a test object in a bioassay. This is due to the fact that the earthworm habitat is limited to organic matter [4]. In addition, the majority of modern methods in the environmental monitoring of soil systems (including rapid assessment tools) focus on the so-called term ‘soil quality’. This term reflects the dynamic features of soil systems, which can often be observed on the soil surface at a depth from 0 to 25 cm [5]. The conventional concept of soil samples and soil testing interpretation, which are based on the internal features of soil, reveal that all assessment methods should involve findings regarding all the changes occurring in the whole-soil profile [6]. There is an assumption that the investigation of the mechanisms of pollutant migration and their leaching into the subsoil should attract more attention from researchers than just the assessment of the pollutant content in the topsoil layer. The whole-soil profile toxicity bioassay could be a preliminary test to identify whether the toxicant has migration mobility or leaching activity. For instance, Verbeeck and co-authors show that the knowledge connected with the mobility and levels of AsO4 in topsoil samples is not enough to describe the vertical transport of AsO4 in the soil profile [7]. In addition, soil profile monitoring is able to provide more detailed and deeper findings of the damage occurring in soil systems in the case of long-term heavy metal mobility and accumulation [8], the wastewater irrigation of bacterial communities in agricultural soils [9] and changing land-use strategies and crop rotations [10].

The lack of data regarding the usage of modern bioassays for full-profile soil testing can be ascribed to the specific limitations of the chosen test organisms. Moreover, rapid bioassay methods are time-consuming activities, and they require a well-equipped laboratory [5]. Meanwhile, enzymes are not needed for the presence of humic organic matter for their operation, unlike living test organisms, which enable enzymes to be used for full-profile soil testing. In addition, enzymes can be used as a biological part of enzyme-based biosensors. Previously, our research group demonstrated the applicability of a bioluminescent enzyme system for the assessment of heavy metal and pesticide levels in soils [3,11,12,13,14]. The bacterial coupled enzyme system consists of two enzymes, namely, NAD(P)H:FMN-oxidoreductase and luciferase (Red + Luc), catalyzing the following reactions (1 and 2): NAD(P)H:FMN -oxidoreductase (Red) 
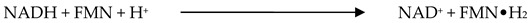
(1)
Luciferase (Luc)
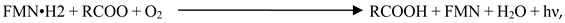
(2)
where FMN and FMN•H_2_ are the oxidized and reduced forms of flavin mononucleotide, NAD^+^ and NADH are the oxidized and reduced forms of nicotinamide adenine dinucleotide (phosphate), RCHO is myristic aldehyde, and RCOOH is the corresponding fatty acid. The pollution of the mixture is determined by a change in the value of the intensity of bioluminescence in the presence of the sample in comparison with the control [15]. At the same time, it should be mentioned that, for the coupled bioluminescent system, signal processing systems were proposed, namely, a compact and portable luminometer, a disposable microfluidic chip and a sampler adapter. These transducers could be used in biosensors for the detection of heavy metals in water samples [16].

Therefore, this paper is devoted to investigating the effects of soil samples with different man-made impacts, namely, agricultural grassland, land treated with remediation processes for 10 years (10-year fallow land) and uncontaminated (virgin) land, on the activity of the Red + Luc enzyme system. This investigation is needed for the identification of possible challenges that could arise during the development and expansion of the analytical application of biosensors for full-profile soil toxicity assessments.

## 2. Materials and Methods

### 2.1. Soil Collection and Characterization

Full-profile soil samples from three experimental zones, namely, agricultural grassland (55°59′57.82″ N, 95°10′14.77″ E), 10-year fallow land (treated with remediation processes for 10 years, 56°0′11.06″ N, 95°10′8.91″ E) and uncontaminated virgin land (56°0′11.15″ N, 95°10′13.47″ E), were collected in the Experimental Production Farm ‘Solyanskoye’ (Rybinskiy Region, Krasnoyarsk Region, Russia). The terrain altitude is 355 m. The soils from ‘Solyanskoye’ are subject to complex erosion (17% of the total farm area) to a small or medium extent. The soils have a high fertility coefficient and natural resource potential equal to 1.87, despite the fact that the land of the experimental production farm is unstable due to ploughing (the coefficient of ecological stability is K = 0.30), and it is considered as highly modified (anthropogenic conversion factor = 6.56) [17]. High doses of mineral fertilizers and pesticides are used in ‘Solyanskoye’; therefore, the agricultural grasslands are contaminated with agrochemicals [18].

Soil sampling, transportation and storage were mentioned in the requirements of the government regulatory documents [19]. Six soil samples from each zone were collected. The samples were taken layer by layer, with one sample from each genetic zone horizon (between 0 and 125 cm). The soils were classified as chernozem of different types [18]. The soils were rationalized, air-dried at room temperature, ground in a mortar, sieved through a sieve with holes of 1 mm and thoroughly mixed.

The properties of the soil samples were determined following the classical agrochemical and physical methods, which are mentioned in the requirements of the government regulatory documents. The acid–base coefficient (pH_KCL_) of the soil was determined using the potentiometric method in KCl solution [20,21]. The percentage of humus levels and labile humic substances (LHSs) in aqueous extracts from the soils was estimated using the Tyurin method. Labile humic substances were extracted with a 0.1 N NaOH solution (1:20 ratio of the soil and solvent). Humic acids (HAs) were precipitated with sulfuric acid. The content of fulvic acid (FA) was calculated from the difference between the total content of LHSs and HA. The HA:FA ratio was calculated to determine the type of humus.

### 2.2. Bioluminescent Enzymatic Assay

Full-profile soil toxicity screening was carried out using the previously developed method of measuring the activity of the Red + Luc system in the presence of soil samples [3]. Aqueous extracts from the soils for the bioluminescent enzymatic assay were prepared using the method described earlier [11]. Briefly, 5 g of soil lot was diluted with 25 mL of H_2_O and then shaken, centrifuged and filtered. 

To determine the impact of the aqueous soil extracts from the soils on the activity of the Red + Luc enzyme system, a reaction mixture of the following composition was used: 300 μL of 0.05 M potassium phosphate buffer (pH = 6.9); 5 μL of enzyme solution; 50 μL of 0.0025% (*v*/*v*) aldehyde solution; 100 μL of 0.4 mM NADH solution; 50 μL of distilled water (control) or test solution; and 10 μL of 0.5 mM FMN solution. A vial of the Red + Luc solution (Laboratory of Nanobiotechnology and Bioluminescence, Institute of Biophysics, Siberian Branch, Russian Academy of Sciences, Krasnoyarsk) contained 0.4 mg/mL of luciferase obtained from a recombinant *E. coli* strain and 0.18 units of activity of NAD(P)H:FMN oxidoreductase from *Vibrio fischeri*. Enzyme solutions were prepared in a 0.05 M potassium phosphate buffer (pH = 6.9).

For the full-profile analysis of the soil samples, the reaction mixture was placed in a luminometer cuvette (GloMax 20/20^n^ Luminometer, Promega, Madison, WI, USA), and the luminescence intensity was measured. The residual luminescence was calculated according to the formula (I/I_0_) × 100%. To determine the impact of the analyzed soil sample, the determination scale described earlier [11,12] was used as follows: at I/I_0_ > 80%, the analyzed soil sample was considered to have no impact, at 50% < I/I_0_ < 80% the analyzed soil sample was considered as having an impact, and at I/I_0_ < 50%, the analyzed soil sample was considered to have a significant impact.

### 2.3. Data Processing

Data are presented as a mean value (M) ± standard deviation (s). Statistical analyses were performed with Statistica v10 (StatSoft Inc., Tulsa, OA, USA). All the measurements were repeated 3 times. The results were considered statistically significant at *p* < 0.05.

## 3. Results

The chemical properties and profile form of the tested soil samples were identified (In Appendix A). It should be noted that the tested soil samples were classified as agricultural grassland, 10-year fallow land (treated with remediation processes for 10 years) and uncontaminated (virgin) land.

Figure 1A presents the influence of the agricultural-grassland samples on the activity of the Red + Luc enzyme system. The values of the residual light intensity of the enzyme system have a low dependence on the sampling depth. The minimum value of the system, which is equal to 24.2 ± 0.9%, was recorded during the assessment of the agricultural-grassland sample with a sampling depth from 0 to 10 cm. The maximum value of the light intensity of the system (63.7 ± 0.3%) was obtained during the agricultural-grassland sampling at a depth from 75 to 120 cm. The level of humus and the content of LHSs in the studied farm field samples decrease from the top to the bottom of the soil profile (Figure 1B,C). The humus, which is located in the topsoil, is characterized as belonging to the fulvic–humic type. At a lower depth, it changes to the humic–fulvic type. The pH_KCL_ value was classified as weakly acidic throughout the full soil profile.

The impact of the 10-year fallow-land samples on the activity of the enzyme system is presented in Figure 2A. The level of the light intensity of the system increases with depth. The light intensity values of the enzyme system, namely, 36.8 ± 3.0% and 99.7 ± 8.9%, were recorded upon testing the soil samples taken at a depth of 0–10 cm and 75–120 cm, respectively. The humic level and the content of LHSs are higher in the topsoil than in the subsoil (Figure 2B,C). The types of humus in the virgin land range from the fulvic–humic type to the humic–fulvic type. The pH_KCL_ value was classified as faintly acidic in the topsoil, but the subsoil sample had a neutral pH_KCL_ value.

The effect of the virgin-land samples on the activity of the Red + Luc enzyme system is shown in Figure 3A. The change in the residual light intensity of the system in the presence of the virgin-land samples occurred in the same manner as in the 10-year fallow-land samples. The residual light intensity of the system equal to 32.7 ± 3.9% was registered for the sample taken at a depth of 0–10 cm. On the contrary, the 95.5 ± 11.7% value of the residual light intensity of the system was recorded for the sample taken at a depth of 75–120 cm. The humic level and the content of LHSs decrease dramatically from the top to the bottom of the soil profile (Figure 3B,C). The types of humus in the virgin land range from the fulvic–humic type to the humic–fulvic type. The pH_KCL_ value of the topsoil was classified as faintly acidic, but the subsoil had a neutral pH_KCL_ value.

## 4. Discussion

The investigation of the impact of twenty-one soil samples collected from three full-profile soil systems (agricultural grassland, 10-year fallow-land (treated with remediation processes for 10 years) and uncontaminated (virgin) land) on the activity of the Red + Luc enzyme system shows that the enzyme system has the possibility of being used as a qualitative tool for the identification of the distribution of pollutants in the whole-soil profile. The inhibition of the enzyme system’s activity by the soil samples decreased with sampling depth: from 36% to 100% and from 39% to 100% for the virgin-land samples and 10-year fallow-land samples, respectively. The low residual light intensity values of the Red + Luc enzyme system during the tests of the soil samples from a depth of 0–30 cm are probably associated with the high humic organic matter content in their topsoil layers (from 6 to 10%) rather than with the presence of a toxic agent.

Our research group previously showed that the accuracy of the bioluminescence assay, developed for and applied in soil toxicity assessments, is related not only to the presence of a toxicant but also to the humic organic matter content [11,12,13,14]. For instance, humic substances at levels higher than 1 mg/L have a negative impact on the substrate concentration in the bioluminescent reaction, which leads to a decrease in the activity of the enzyme system. This means that the soil samples with a high content of humic substances (more than 1 mg/L), which are also unpolluted, could be falsely classified as toxic by bioluminescent bioassays [22,23]. In our case, the enzymatic bioassay showed that the humic content in the soil samples within a range from 0.77 to 1.41% and LHSs with no more than 50 mgC/100 g did not cause any inhibition of the enzymatic activity.

The consideration of the data regarding possible the reasons for the light intensity variations in the enzyme system during the soil profile assessment of the agricultural-grassland samples leads to the conclusion that the intensive use of soils in agriculture results in their contamination not only in the topsoil layer (sampling depth from 0 to 20 cm) but also in the full-profile layers. This conclusion is supported by the results of the enzymatic bioassay of the full-profile agricultural-grassland samples showing that, despite the decrease in the humic organic matter content in the samples with the sampling depth, which is one of the interfering factors for the enzymatic bioluminescent bioassay, the samples were classified as toxic and highly toxic when the values of the residual light intensity of the enzyme system were not higher than 60% [11]. The high inhibition levels of the activity of the Red + Luc enzyme system are probably caused by mineral fertilizers and pesticides, which are commonly used in farm fields for crop production, as well as in the ‘Solyanskoye’ farm [18]. This supplementary chemical usage may be confirmed by the acidic pH level (pH_KCL_ = 4.5) of the subsoil layers of the farm field samples, which is not typical. The acidic pH_KCL_ level of the subsoil layers is connected to sulfate accretion. Accretion has a negative effect on plant growth and development and alters the balance of available soil elements for plants [24].

Moreover, we suggest that the results of the virgin-land full-profile contamination bioassay could be the basis for involving natural reference soil samples while applying rapid bioassay methods to assess the contamination of complex matrices. It should be noted that rapid bioassay methods are usually criticized due to the use of questionable samples as references [3]. As for the rapid bioassay assessment of soil toxicity, the frequently used and commercially available artificial reference soil samples often represent the mineral composition of soils [25,26], but it is still unclear whether the mineral composition is properly reflective of the biological, chemical and physical composition of soils. In this case, the virgin land, firstly, is a native one to soil biodiversity in the territory of the Krasnoyarsk Region, Russia. Secondly, this land is composed of conditionally clean soil that has not experienced any man-made impact. As such, the results concerning the impact of the virgin-land samples, collected in the Krasnoyarsk Region, on the activity of the Red + Luc enzyme system show that the studied soil samples better meet the requirements for reference soil samples, which are to be used to develop and validate the enzyme-inhibition-based assay for the prediction of the toxicity of pollutants in agroecosystems of the Krasnoyarsk Region rather than commercially available artificial reference soil samples.

Thus, for the successful commercialization of such biosensors, it is necessary to develop biosensor software to match the humic organic matter content contribution to the Red + Luc enzyme system’s activity in the presence of a tested soil. This software might be similar to the algorithm developed based on 51 non-commercial standard soil samples for the identification of their inhibitory effects on enzyme systems [12]. That is, the accuracy of the bioluminescent biosensors might significantly be improved by taking into account the interfering effect of humic organic substances in a tested soil sample.

## 5. Conclusions

The full-profile analysis of the agricultural-grassland samples showed an inhibitory effect on the enzymes of luminescent bacteria, which suggests that there is toxicant migration throughout the whole-soil profile in soils involved in crop production. This assumption is based on the inhibitory effect of the full profile of 10-year fallow-land and virgin-land samples on the activity of the Red + Luc enzyme system, which decreases with depth. Such an inhibitory effect is connected to a decline in the humic organic matter content in the tested soil samples. Thus, the bioluminescent enzymatic method is suitable for revealing the potential contamination of full-profile soil samples, and it can be used as the basis for new methods and biosensors to screen contaminant migration in soil profiles. At the same time, software to match the standard enzymatic activity of enzyme-based biosensors in a transducer could justify a consuming capacity of the developed bioluminescent-enzyme-based biosensors for full-profile soil toxicity screening.

## Figures and Tables

**Figure 1 biosensors-12-00353-f001:**
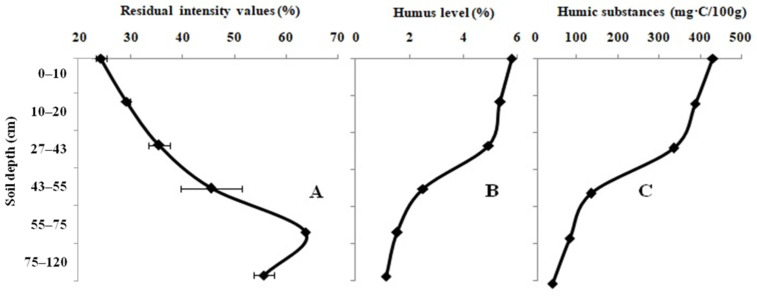
Variation in the residual light intensity of the Red + Luc enzyme system (**A**); humic (**B**) and labile humic substances (**C**) in the full profile of the agricultural-grassland samples.

**Figure 2 biosensors-12-00353-f002:**
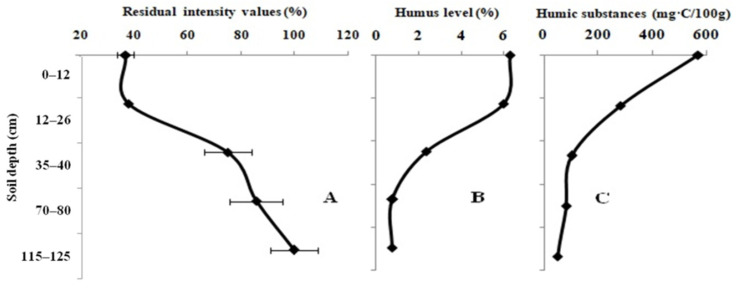
Variation in the residual light intensity of the Red + Luc enzyme system (**A**); humic (**B**) and labile humic substances (**C**) in the full-profile of the 10-year fallow-land samples.

**Figure 3 biosensors-12-00353-f003:**
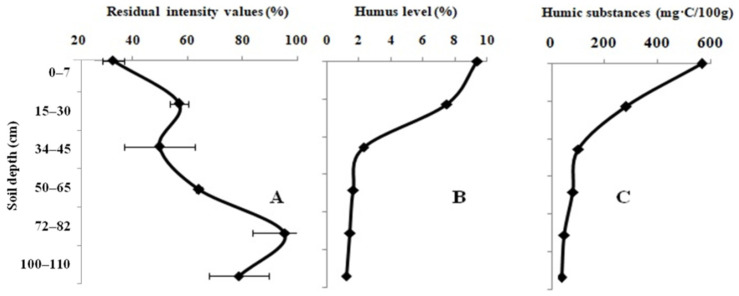
Variation in the residual light intensity of the Red + Luc enzyme system (**A**); humic (**B**) and labile humic substances (**C**) in the full-profile of the virgin-land samples.

## Data Availability

Not applicable.

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
