# Peer review of "Bioluminescent-Inhibition-Based Biosensor for Full-Profile Soil Contamination Assessment"

_biosensors, 2022, doi:10.3390/bios12050353_

Round 1

Reviewer 1 Report

The authors of the manuscript “Bioluminescent inhibition-based biosensor for full-profile soil contamination assessment” tested three types of soil samples using Red-Luc enzyme. Their results showed that this method can be used to detect soil pollution. The significance of this manuscript is clear and its potential applications could be beneficial for agriculture.

I have a few questions.

1: since soil samples contain more complex matrics than water, are there any other contents that should be considered besides humus?

2: are reactions 1 and 2 correctly presented?

3: is there any reason that the determination scale was determined as in the last sentence on Page 3?

Reviewer 2 Report

In the reported manuscript, Valentina and co-workers are reporting a Bioluminescent based method for analyzing contaminations in the soil. The current manuscript is in a very crude format and it does not specify the importance of the findings. Specifically, this manuscript does not specify, what type of contaminations will be detected by method. Without any specificity, it is very difficult to assess the significance of this work. methods and the results are very poorly discussed in the manuscript. These sections must be significantly improved prior to submitting to any other journal. There are grammatical and textual errors which needs to be revised carefully prior to re-submission to a different journal. I will not be able to recommend this manuscript due to the lack of scientific soundness of this submitted version. However, I suggest authors to revise this manuscript carefully and attempt to re-submit to a more specific geology/geochemistry related journal. The content of this manuscript may not meet the significance/relevance to be published in Biosensors.

In addition, following changes are recommended:

(1). Abstract must be revised thoroughly. Authors should be specific and quantitative in the abstract to highlight the significance of the findings.

(2). In the introduction section, authors must provide more descriptive background information with sufficient references. The current introduction is very poor.

(3). Authors must specify what type of luminous bacteria they are discussing in the introduction and should provide scientific names.

(4). Specify the chromophore and explain the luminescence process by using a cartoon scheme for the clarity. The current representation format is very crude.

(5). Authors should tabulate and compare the enzyme based assays they are discussing in the introduction section. 

(6). Specifically, what type of agrochemicals authors are detecting here?

(7). What is the specificity, sensitivity and the selectivity of this assay method?

(8). All samples preparation methods should be describe thoroughly to clarify the testing method.

(9). How did authors measures the luminescence signal? can authors provide the spectral profile? What is the specific wavelength used for this detection method? These information are critical to discuss.

(10). Provide a plausible mechanism with the sue of representative chemical contaminant structure to understand how the detection method function.

(11). Authors should discuss clearly why the relative signal intensity change very unusually as a function of depth in all reported data. Did authors replicate these analysis data?

(12). Conclusion section is very poor and should revised thoroughly.

Round 2

Reviewer 2 Report

I recommend to accept in the current form. after a final grammar check.